# Field-Driven Magnetic Phase Diagram and Vortex Stability in Fe Nanometric Square Prisms

**DOI:** 10.3390/nano12234243

**Published:** 2022-11-29

**Authors:** Mauricio Galvis, Fredy Mesa, Johans Restrepo

**Affiliations:** 1Group of Magnetism and Simulation G+, Institute of Physics, University of Antioquia, A.A. 1226, Medellín 050010, Colombia; 2NanoTech Group, Facultad de Ingeniería y Ciencias Básicas, Fundación Universitaria Los Libertadores, Cra. 16 No. 63a-68, Bogotá 111221, Colombia

**Keywords:** iron nanoprisms, micromagnetics, vortex states, magnetic phase diagram, aspect ratio

## Abstract

In this work, we deal with the zero temperature hysteretic properties of iron (Fe) quadrangular nanoprisms and the size conditions underlying magnetic vortex states formation. Different aspect ratios of a square base prism of thickness *t* with free boundary conditions were considered in order to summarize our results in a proposal of a field-driven magnetic phase diagram where such vortex states are stable along the hysteresis loops. To do that, a Hamiltonian consisting of exchange, magnetostatic, Zeeman and cubic anisotropy energies was considered. The time dynamics at each magnetic field step was performed by solving the time-dependent Landau–Lifshitz–Gilbert differential equation. The micromagnetic simulations were performed using the Ubermag package based on the Object Oriented Micromagnetic Framework (OOMMF). Circular magnetic textures were also characterized by means of topological charge calculations. The aspect ratio dependencies of the coercive force, nucleation and annihilation fields are also analyzed. Computations agree with related experimental observations and other micromagnetic calculations.

## 1. Introduction

Both ferromagnetic ultrathin films, and more general quadrangular prisms, offer a remarkable chance for exploring the interplay between magnetism and topological features such as the aspect ratio or the geometric shape. In this respect, iron (Fe) has been the reference material par excellence, and many works have been devoted to studying its magnetization reversal mechanisms under different conditions [1,2,3,4,5,6,7]. In particular, the study of vortex-state magnetic textures in soft ferromagnetic structures [8,9,10], mainly driven by the magnetostatic energy, which favors magnetic closure structures with in-plane magnetization except at the core (magnetic soliton), is of considerable current interest as long as they do not produce stray magnetic fields with a cost of course in the exchange energy. In fact, the delicate balance between the different energies (e.g., exchange, Zeeman, magnetostatic, and anisotropy) can also give rise to the formation of a vortex core with a high exchange energy density near the vortex center with an out-of-plane magnetization as a mechanism to decrease the exchange energy. This is an important feature, for instance in patterned nanomagnets [9], where dipolar interactions are not desirable for specific applications [11] such as spin-transfer torque nano-oscillators, high-density magnetic data storage, magnetic field sensors, logic operation devices, spintronics, biomedical applications [12,13,14], etc. In this way, a full characterization of this type of magnetic textures, not only experimentally but also from a computational standpoint, is needed. Both square and circular structures, among others such as elongated iron nanodots [15], can exhibit vortex states or even double-vortex nucleation during magnetization reversal as it occurs in circular iron nanodots [16]. However, in the latter, differently from square structures, vortexes are characterized by a continuous and smooth varying magnetization, different from the Landau triangular domain structures in square films, which have been less studied [9].

Some micromagnetic works on (Fe) have also been published recently in the rock magnetism, paleomagnetism, meteorites and planetary magnetism literature, where the stability of the different phases plays an important role [17,18]. In particular, paleomagnetic studies offer an insight about the strength of magnetic fields present during the evolution of the solar system where particles’ remanences can be stable for billions of years. Hence, the importance of characterizing the different magnetic states. Within this thematic line, the magnetic stability of flower versus single vortex states was studied in cubic (Fe) particles as a function of cube edge length as well as the aspect ratio dependence of the threshold lengths where individual elongated iron particles transit from a magnetically uniform single domain to a non-uniform multidomain state [19]. Thermal and temporal stability has also been studied for small cubic and spherical iron grains above 25 nm of paleomagnetic importance [20]. These studies, however, do not explore the hysteretic properties of the iron particles.

The mechanical stability of vortex states when an external magnetic field is applied is also interesting. An example of single vortex states that can arise when considering more complex and realistic particle geometries such as those in ellipsoidal iron particles is nicely illustrated in [21], where the orientation of a vortex is determined mainly by the ellipsoidal geometry. More concretely, a vortex core is parallel to the major axis for prolate ellipsoids and parallel to the minor axis for oblate ellipsoids. This work is one of the scarce studies where hysteresis loops were modeled by using micromagnetic simulations for pure iron particles, but it was restricted to very specific shapes and geometries. Some other important efforts [7,8,9,10] have also been made in simulating hysteresis loops of other ferromagnetic structures, mainly Permalloy, exhibiting curling states in square-shaped structures to understand the associated vortex dynamics such as vortex core displacement with a uniform in-plane applied field, apart from other parameters such as the nucleation and annihilation fields of these structures or the zero-field susceptibility as a response function of the ease with which the vortex core can be displaced by the field. This last one is a measure of the demagnetizing process and the degree of reversibility in the central region of the hysteresis loops, which is a key feature for paelointensity studies.

Thus, in this work, we investigate systematically, from a micromagnetic point of view at zero temperature, the conditions favoring vortices’ formation in iron quadrangular nanoprisms and their stability under the influence of a unidirectional and uniform in-plane applied field. Different dimensions are considered, in order to scrutinize the influence of geometry, and more specifically, the aspect ratio. Our results are finally summarized in a proposal of a field-driven magnetic phase diagram. It must be stressed that such prism-type geometries have been even experimentally synthesized [22].

## 2. Materials and Methods

We consider a square prism L×L×t by using free boundary conditions (f.b.c) where *L* is the cubic sample edge length along *x* and *y* axes with −L/2≤x,y≤L/2, whereas *t* is the thickness or height of the prism along the *z* axis with t>0. In this work, *L* ranged between 30 and 300 nm, whereas *t* was taken to vary between 3 and 120 nm in order to consider a wide spectrum of aspect ratios (t/L). The size of the discretization cell edge for sample meshing was set up at 3 nm, which is smaller than the magnetic exchange length of iron where $l_{ex}=\sqrt{2A/(\mu_{0}M_{s}^{2})}=3.4$ nm, being $A=2.1\times 10^{-11}$ Jm the exchange stiffness constant, $M_{s}=1.7\times 10^{6}$ A/m the saturation magnetization and $\mu_{0}$ the permeability of free space [23]. Simulation cell size is an important factor to obtain feasible results in order to make the relative angular deviation between two magnetization vectors of neighboring simulation cells small enough, which is governed by exchange length [24,25]. Additionally, a cubic anisotropy constant $K=4.8\times 10^{4}$ J/m${}^{3}$ was considered [15,16].

The Hamiltonian of our system reads as follows:(1)$$\begin{array}{cc} H= & -Am\cdot \nabla^{2}m-K[{\left(m\cdot u_{1}\right)}^{2}{\left(m\cdot u_{2}\right)}^{2}+{\left(m\cdot u_{2}\right)}^{2}{\left(m\cdot u_{3}\right)}^{2}\\ \; & +{\left(m\cdot u_{1}\right)}^{2}{\left(m\cdot u_{3}\right)}^{2}]-\frac{1}{2}\mu_{0}M_{s}m\cdot H_{d}-\mu_{0}M_{s}m\cdot H \end{array}$$
where $u_{1}$, $u_{2}$ and $u_{3}$ are the unit vectors along the [1,0,0], [0,1,0] and [0,0,1] directions, respectively, $m=M/M_{s}$ is the normalized magnetization, $H_{d}$ is the demagnetizing field caused by magnetostatic interaction, and H is a uniform external field applied along the [1,0,0] easy axis. Hysteresis loops were obtained for $H=H\hat{x}$ along the easy *x*-axis and varying in a wide range of values in order to guarantee saturation at the maximum values.

The dynamics of the magnetization field m is ruled by the Landau–Lifshitz–Gilbert (LLG) equation [26,27,28], which consists of precession and damping terms:(2)$$\frac{dm}{dt}=-\gamma_{0}\left(m\times H_{eff}\right)+\alpha \left(m\times \frac{dm}{dt}\right)$$
or
(3)$$\frac{dm}{dt}=-\frac{\gamma_{0}}{1+\alpha^{2}}\left(m\times H_{eff}\right)-\frac{\gamma_{0}\alpha }{1+\alpha^{2}}m\times \left(m\times H_{eff}\right)$$
where $\gamma_{0}=\mu_{0}\gamma =2.211\times 10^{5}$ mA${}^{-1}$s${}^{-1}$ is the gyromagnetic ratio, α=1 is the Gilbert damping, and:(4)$$H_{eff}=-\frac{1}{\mu_{0}M_{s}}\frac{\delta w(m)}{\delta m}$$

**H**${}_{eff}$ is the effective field where *w* is the total energy density functional of the system given by the sum of the different energy contributions. We want to stress that the election of α=1, instead of a smaller value, does not imply a loss of generality nor does it affect the static pseudo-equilibrium properties [29], and it allows to reduce computing time. Such a value is the one for which the prefactors for precession ($\gamma_{0}/\left(1+\alpha^{2}\right)$) and damping ($\gamma_{0}\alpha /\left(1+\alpha^{2}\right)$) are equal, making these features equiprobable.

Regarding the time evolution at every single *H* step value during the loop, the system was driven during 5 ns along 10 time steps.

The characterization of vortex states was carried out by implementing the calculation of the 2D topological charge (also known as the skyrmionic charge) [30,31,32]:(5)$$Q=\frac{1}{4\pi }\int m\cdot \left(\frac{\partial m}{\partial x}\times \frac{\partial m}{\partial y}\right)dxdy$$

Micromagnetic simulations were performed by using the Ubermag Python ecosystem [33] based on the Object Oriented MicroMagnetic Framework (OOMMF) as micromagnetic calculator [34].

## 3. Results and Discussion

Figure 1 shows some selected hysteresis loops of the magnetization component projected along the *H*-direction (*x*-axis) for a sample with L=300 nm and three different thicknesses, namely t=21 nm, t=27 nm and t=42 nm.

With increasing the height of the prism, the coercive force $H_{c}$ becomes smaller, passing through a maximum, until it finally vanishes at a certain value around t=25 nm. This can be better observed in Figure 2 with black solid circles, which follow the typical trend observed in systems of ferromagnetic nanoparticles where with a decrease of the size of particles, $H_{c}$ increases and then reaches a maximum value at some critical size due to coherent rotation of the magnetization (below around t=12 nm in Figure 2), passing from a multidomain behavior (b) to a pseudo-single-domain behavior (a) as *t* decreases. In these regions (a) and (b), due to the free boundary conditions, magnetization is not uniform in the corners, leading actually to the formation of flower states [19,35]. In these two regions, labeled simply as ${\mathrm{FM}}_{1}$, hysteresis loops are characterized by a high degree of squareness typical of a collinear ferromagnetic behavior, and no vortex states are observed along the cycle.

On the other hand, for a high enough thickness *t* (from around t=21 nm in this case) and as we move from right to left along the corresponding decreasing field branch, vortex (V) states at the inner region of the prism appear for the first time in those linear regions of the M−H loop in Figure 1 where the vortex nucleation field $H_{n}$ is preceded by a sort of activation field (“A” in Figure 1), which triggers domain wall mobility (first discontinuity). This new feature gives rise to another phase labeled as (${\mathrm{FM}}_{1}+V_{1}$) where coercivity still persists. The value of the thickness for which the transition ${\mathrm{FM}}_{1}\to \left({\mathrm{FM}}_{1}+V_{1}\right)$ takes place is denoted by $t_{1c}$ (see Figure 2). As we increase even more the thickness, and we pass for instance from t=21 nm to t=27 nm in Figure 1, there is an intermediate thickness, labeled as $t_{2c}$, above which coercive force is null, and the central region of the cycle is dominated by a well-defined M−H linear relationship passing through zero. This new phase without coercivity ($H_{c}=0$) is labeled as $V_{1}$ phase. Thus, $t_{2c}$ deals with the transition $\left({\mathrm{FM}}_{1}+V_{1}\right)\to V_{1}$. The range or amplitude of the linear region in the hysteresis loop is a measure of the degree of stability of such vortices with the field, and it corresponds to the difference between the nucleation field ($H_{n}$) where the vortex state is observed for the first time, and the annihilation field ($H_{an}$), beyond which the vortex disappears. These values are also plotted in Figure 2 as a function of *t*.

To illustrate this, some examples of magnetic configurations or snapshots, and more concretely, those related with the formation of inner vortex states and their dynamics, are shown in Figure 3. They correspond to the plane located at the half height of the sample (z=t/2) for some selected thicknesses and field values along the decreasing field branch for L=30 nm. As can be observed, as the field is decreased, the vortex core describes a trajectory in the (x,y)-plane, which can be downwards or upwards (i.e., $-\hat{y}$ or $+\hat{y}$ directions, respectively) depending on the configuration geometry of the state adjacent to the nucleation field which is energetically degenerated. This last key feature is evidenced in Figure 3a,b for t=42 nm and t=45 nm, respectively, where, depending on the spatial distribution of the majority region of moments, a vortex core can appear above or below the line y=0 (see Figure 3d,e respectively) as the field is decreased from 50 to 25 mT in the *x*-direction. Once the vortex state is established, its centroid is defined by a linear relationship of the form $y\left(H_{x}\right)=\pm \left(\partial y/\partial H_{x}\right)H_{x}$, where sign election depends on the prevalence of the majority region or domain at the nucleation state. Similar analysis is valid for other aspect ratios.

On the other hand, singularity at the center of the vortices traduces in an out-of-plane magnetized core as a mechanism of reducing exchange energy as is shown in Figure 4a for t=48 nm where vortex nucleation is accompanied by a strong reduction at around 55 mT. The sharpness of this jump suggests that the appearance of vortex states takes place in a critical fashion. Something analogous occurs for the annihilation field when the magnetization goes from being out-of-plane to in-plane. In order to find out the role played by the other energy contributions during vortex formation, these were also computed along the cycle as it is shown in the same figure. As observed, the largest absolute energy change, associated to the formation or annihilation of vortices, comes from the Zeeman energy (see Figure 4d) followed by the magnetostatic energy represented by the demagnetizing energy (Figure 4b) and finally by the exchange and anisotropy energies, which are of the same order of magnitude. The observed changes mean that all the energies are involved in the formation of the vortices but with different intensities.

We want to stress that both H−t profiles and the critical thickness $t_{c}$, above which vortices appear, are in fact strongly influenced by the aspect ratio and the size of the sample. This can be better noticed in Figure 5 for L=45 nm, where, differently from L=300 nm, there is a new transition from region (d) to (e), where the system transits from a phase characterized by vortices with zero coercivity, namely $V_{1}$, to a new mixed phase having non-zero coercivity, labeled as (${\mathrm{FM}}_{2}+V_{2}$), where vortices now can appear oriented along the main axis, i.e., perpendicular to the previous vortex states $V_{1}$. Such a transition takes place at a critical thickness $t_{3c}$, so there is a *t*-driven evolution of the hysteresis loops as *t* increases. This evolution is revealed in Figure 6 where representative cycles are shown for the last four regions from (b)→(e).

Hysteresis loops exhibiting V-states, such as those observed in Figure 1b,c and Figure 6c, are characterized by jumps along the cycle ascribed to discrete domain wall motions and a magnetization reversal mechanism mediated by the formation of such states. Starting from a positive saturated state, as the applied field is decreased, the moments configuration evolves until a vortex is formed at some nucleation field $H_{n}$ where the magnetization drops off sharply. Afterwards, a constant susceptibility region is reached, i.e., the $M_{H}$ dependence is linear, where the vortex core is shifted in a reversible fashion and in such a way that the contribution of the magnetization parallel to the field increases. This process occurs until the vortex state vanishes at some annihilation field $H_{an}$ and then the negative saturated state is again attained. This reversible region, which divides the hysteresis loop into two separated open and irreversible regions, has been experimentally observed in epitaxial Fe layers grown on Si(0 0 1) [7] and also in some micromagnetic studies [8,9,10,16].

As concerns Figure 6b,c, where vortex states are present with and without coercivity, respectively, we perform calculations of the topological charge given by the winding number in Equation (Equation 5). Differently from skyrmions, topological charge values of the order of 0.5 are typical of vortex states with a high degree of in-plane circularity [9,32,36]. The corresponding *H*-dependence of topological charges for these two cases are presented in Figure 7. Thus, it is clear that vortices can dwell in M−H cycles regardless of the coercivity and the width of the regions where the topological charge is different from zero and close to 0.5 is a measure of the stability of those magnetic states. Such regions correspond to those of constant magnetic susceptibility where the relationship between *M* and *H* is linear.

An interesting feature occurs when passing from region (d)→(e) in Figure 5, or equivalently when passing from (c)→(d) in Figure 6. It should be noted that at this stage, the aspect ratio t/L is close to one, so that the prism changes from having a flattened shape to an elongated one along the *z*-axis. This geometric feature has a deep influence upon the involved energies, more concretely, on the demagnetizing energy, which is the one involving the geometric shape of the sample since its origin is magnetic dipolar in nature and in which the long-range magnetic interactions between the different discretization cells are taken into account, thus determining domains formation. Likewise, as the sample becomes thicker, the cubic character of the magnetocrystalline anisotropy makes it so that the associated energy is also minimized along the *z*-direction, which is one of the easy magnetization axes. Thus, the interplay between geometry and energies competition gives rise to a spatial reorientation of the V-states in which now they dwell along the yz planes instead of along the xy planes. In other words, there is a geometric transition in which the system goes from having vortices dwelling along the xy-plane to vortices along the yz-plane. This transition produces the elongated and open hysteresis loops observed in Figure 6 but with some marked degree of narrowness in the center. Since the system at this stage is saturable with non-zero coercivity, we have labeled this phase as ${\mathrm{FM}}_{2}+V_{2}$. This fact was verified through direct observation of the magnetic configurations at the midplanes, i.e., x=0, y=0 and z=t/2 for the different M−H curves.

The same analysis was carried out for a wide range of (*t*,*L*) pairs, which allows us to summarize our results in a proposal of field-driven magnetic phase diagram shown in Figure 8. Thus, in the range of nanometric dimensions, four different phases can be identified, which are labeled as ${\mathrm{FM}}_{1}$, ${\mathrm{FM}}_{1}+V_{1}$, $V_{1}$ and ${\mathrm{FM}}_{2}+V_{2}$. This last one dominates the region in the phase diagram where t/L>1, whereas the first three take place for t/L<1. This means that the aspect ratio plays an important role in determining the magnetic properties of the system. Summarizing, the ${\mathrm{FM}}_{1}$ phase is characterized by hysteresis loops with a high degree of squareness with non-zero coercivity where magnetization reversal occurs in a coherent fashion and the system behaves magnetically soft. In this region, prisms are geometrically flattened. As we progressively increase the thickness of the sample, a vortex state emerges, evolves and it becomes more stable until the zero coercivity $V_{1}$ phase is established, passing through a non-vanishing coercivity mixed state ${\mathrm{FM}}_{1}+V_{1}$. Here, vortex states at the midplanes of the sample inhabit the xy planes parallel to the substrate of the film. Finally, as the aspect ratio approaches one and the thickness becomes increasingly relevant, vortices undergo a spatial reorientation toward the yz planes perpendicular to the basal plane of the prism giving rise to the onset of the ${\mathrm{FM}}_{2}+V_{2}$ phase. In this phase, the system behaves magnetically harder, it is more difficult to saturate, the loops are elongated, and the magnetization switching mechanism is gradual.

## 4. Conclusions

The hysteretic properties of iron prisms at nanometric scale and the close interdependence with the aspect ratio and the conditions underlying magnetic vortex states formation have been addressed by means of micromagnetic calculations. Results were summarized in a proposal of a field-driven magnetic phase diagram where such vortex states are stable along the M−H curves and where four different phases were identified. We are confident that our results, which are assumed to be similar for other ferromagnetic and nanometric prisms, could be important for the design of devices of technological importance, such as spin valves, where the magnetic configurations at applied static fields, can influence, for example, the electrical transport properties and magnetoresistance.

Finally, as for exploring applied field in other directions (e.g., the *y* or *z*-axis), field orientation-dependent magnetization reversal in sub-100 nm iron structures is currently under progress. In this case, and due to the symmetries involved, we expect a threshold in the aspect ratio *t*/*L*, for values greater than one, where a field applied along the *z*-axis will favor the formation of vortices coplanar to the *y*-*z* or *x*-*z* crystallographic planes, giving rise to similar magnetic states such as the ones presented here. It has been already demonstrated that the shape and crystallographic orientation of a single particle rules the orientation of its vortex core.

## Figures and Tables

**Figure 1 nanomaterials-12-04243-f001:**
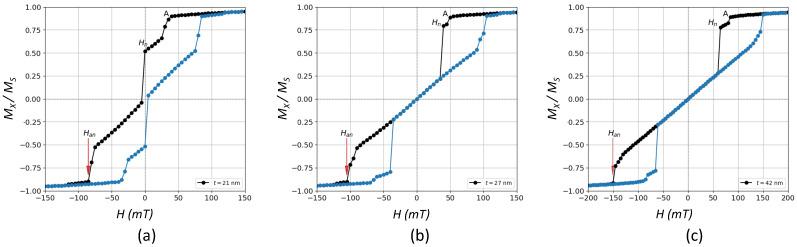
Zero−temperature hysteresis loops for samples with L=300 nm and three different thicknesses, namely **(a**) t=21 nm, (**b**) t=27 nm, and (**c**) t=42 nm. Different colors are used to differentiate the branches corresponding to the decreasing field branch (black circles) and the increasing field branch (blue circles). Nucleation ($H_{n}$) and annihilation ($H_{an}$) fields are indicated. Coercive force ($H_{c}$), only different from zero in (**a**), corresponds to the field for the magnetization passing through zero.“A” stands for a first activation field ascribed to a high mobility of domain walls before vortex nucleation.

**Figure 2 nanomaterials-12-04243-f002:**
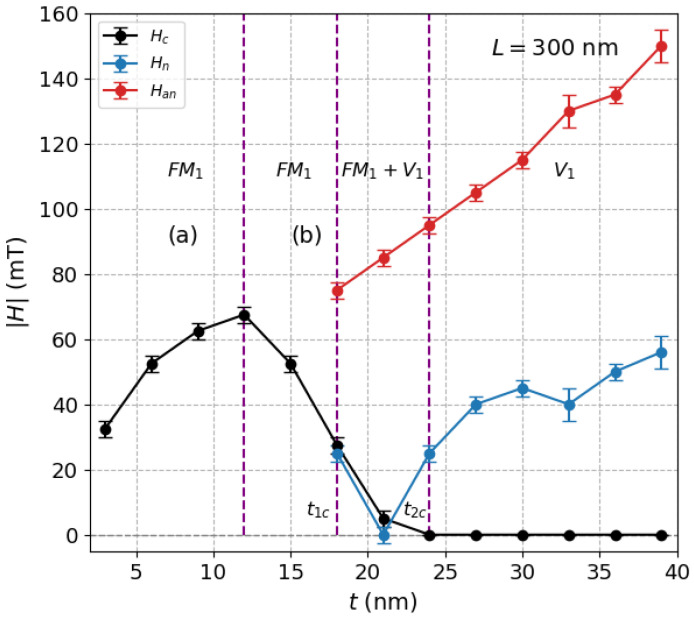
Thickness dependence of the absolute value of coercivity ($H_{c}$), nucleation field ($H_{n}$) and annihilation field ($H_{an}$) for prisms with L=300 nm. This plot shows the regions corresponding to the phases ${\mathrm{FM}}_{1}$, (${\mathrm{FM}}_{1}+V_{1}$), the pure vortex states $V_{1}$, and the respective transition thickness values $t_{1c}$ and $t_{2c}$.

**Figure 3 nanomaterials-12-04243-f003:**
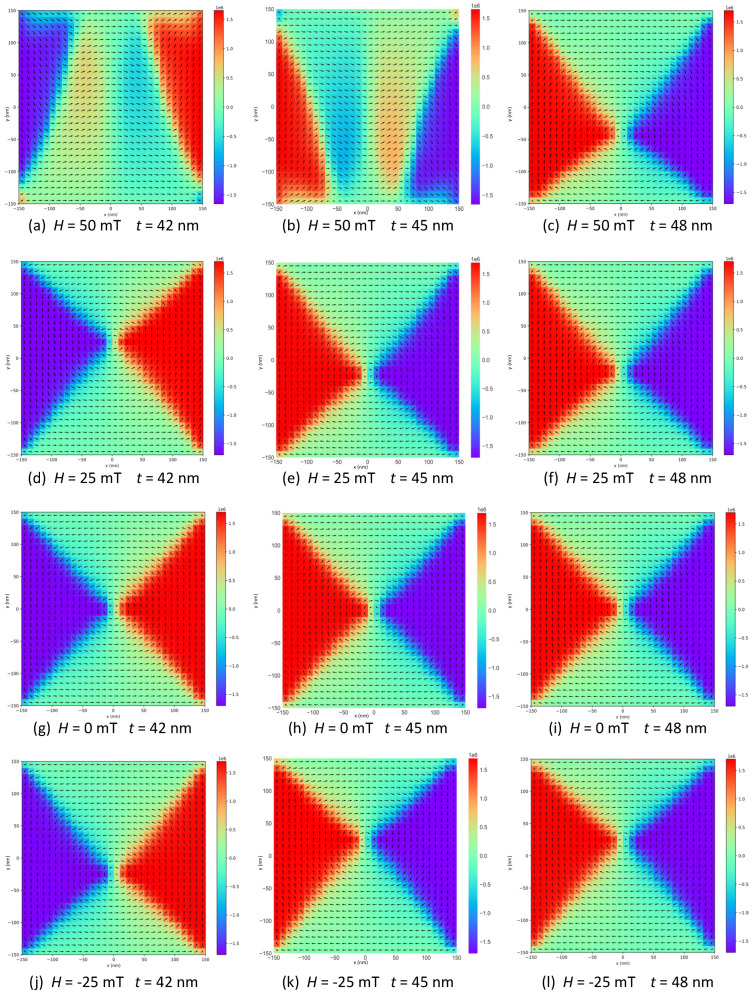
Magnetic configurations for three different thicknesses, namely t={42,45,48} nm (rows), L=30 nm and four different field values, namely H={50,25,0,−25} mT (columns).

**Figure 4 nanomaterials-12-04243-f004:**
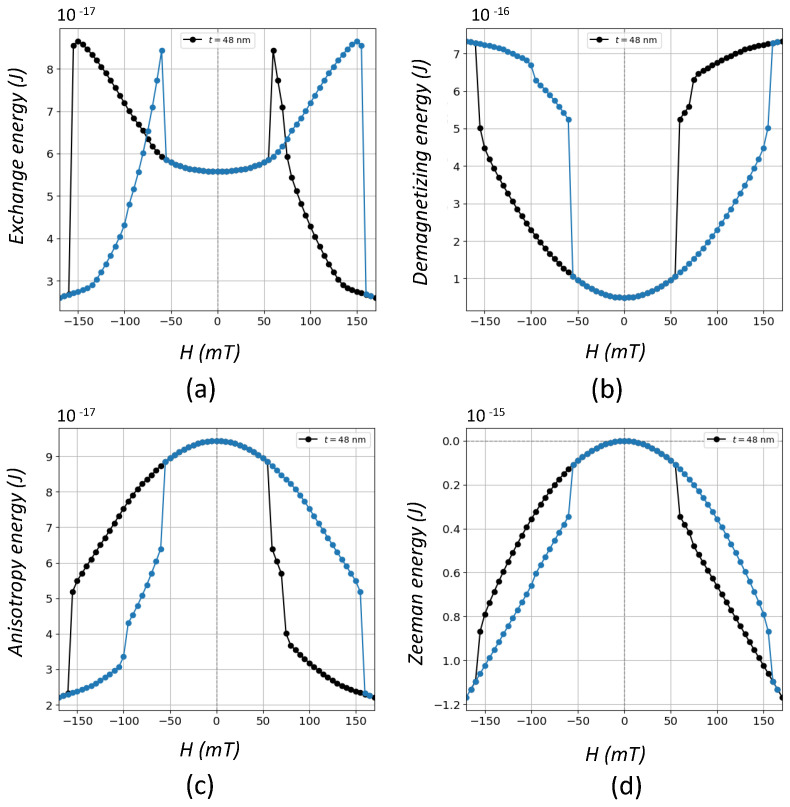
Energy cycles for a prism with L=300 nm and t=48 nm for (**a**) Exchange energy, (**b**) Demagnetizing energy, (**c**) Anisotropy energy and (**d**) Zeeman energy. Two colors are used to differentiate the branches corresponding to the decreasing field branch (black circles) and the increasing field branch (blue circles).

**Figure 5 nanomaterials-12-04243-f005:**
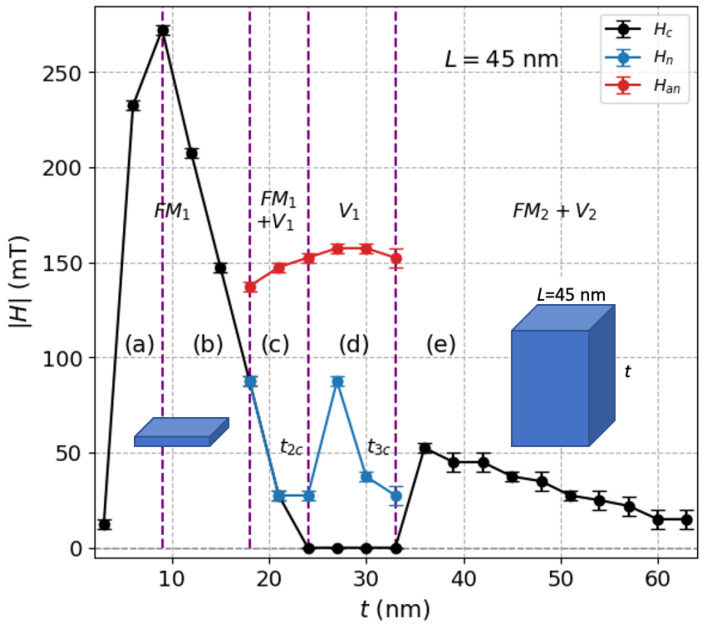
Thickness dependence of the absolute value of coercivity ($H_{c}$), nucleation field ($H_{n}$) and annihilation field ($H_{an}$) of $V_{1}$-states for prisms with L=45 nm. A non-zero coercivity new phase (${\mathrm{FM}}_{2}+V_{2}$) is observed beyond a critical thickness $t_{3c}$.

**Figure 6 nanomaterials-12-04243-f006:**
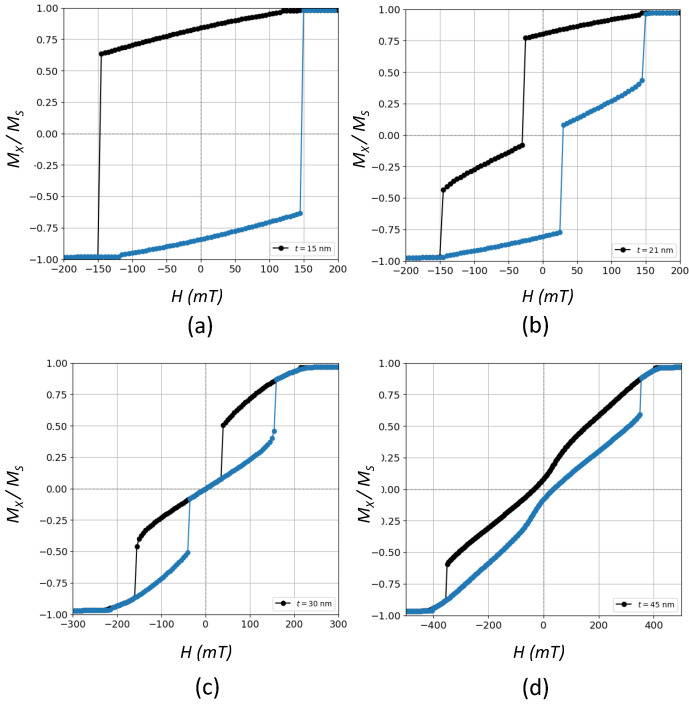
Zero temperature hysteresis loops for prisms with L=45 nm and for (**a**) t=15 nm, (**b**) t=21 nm, (**c**) t=30 nm, and (**d**) t=45 nm, corresponding to the regions (b), (c), (d) and (e), respectively, in Figure 5. Black circles stand for the decreasing field branch and blue circles stand for the increasing field branch. Here, aspect ratio t/L≤1.

**Figure 7 nanomaterials-12-04243-f007:**
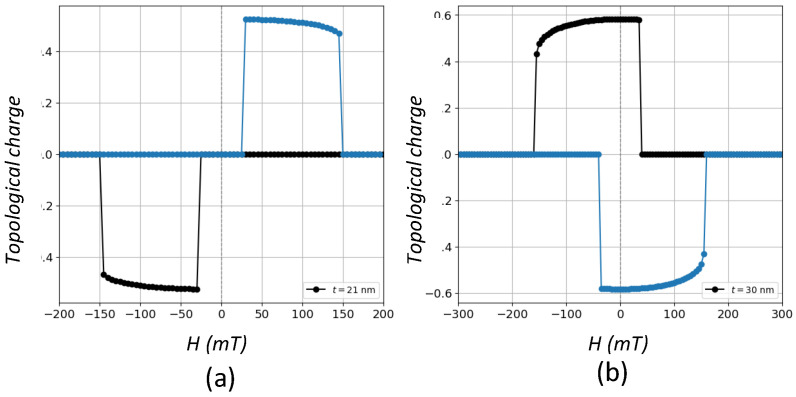
Field dependence of the topological charge for a prism with L=45 nm for (**a**) t=21 nm and (**b**) t=30 nm. Regions where values are close to 0.5 are those where the vortices dwell. Black circles stand for the decreasing field branch of the hysteresis loop and blue circles stand for the increasing field branch.

**Figure 8 nanomaterials-12-04243-f008:**
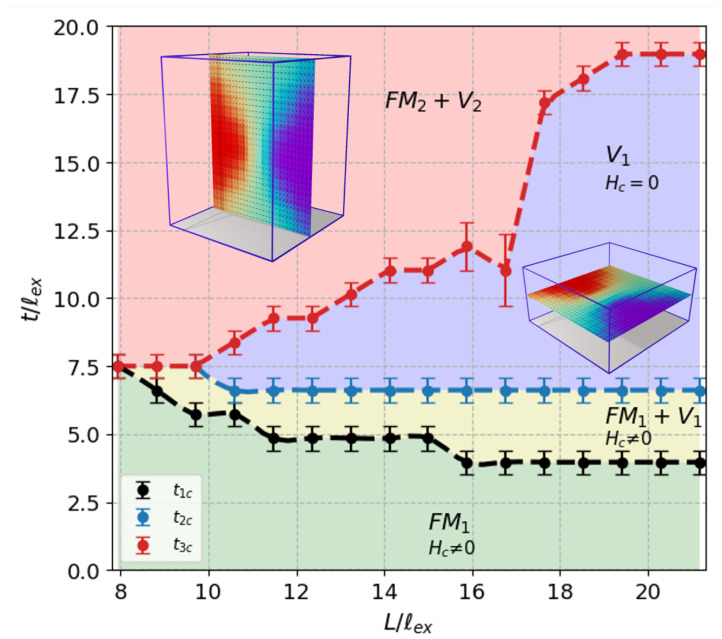
Field−driven magnetic phase diagram *t* vs. *L* in reduced units of the exchange length for an in-plane field applied along the *x*-direction. Four different phases can be identified, which are labeled as ${\mathrm{FM}}_{1}$, ${\mathrm{FM}}_{1}+V_{1}$, $V_{1}$ and ${\mathrm{FM}}_{2}+V_{2}$. The typical shapes of the M−H curves associated with these phases are those shown in Figure 1 and Figure 6, respectively.

## Data Availability

Not applicable.

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
