# Peer review of "Field-Driven Magnetic Phase Diagram and Vortex Stability in Fe Nanometric Square Prisms"

_nanomaterials, 2022, doi:10.3390/nano12234243_

Round 1

Reviewer 1 Report

In the paper, the authors present a study on the vortices formation in iron quadrangular nanoprisms and their stability under the influence of a unidirectional and uniform in-plane applied field, in which different aspect ratios of a square base prism of thickness with free boundary conditions were considered. The aspect ratio dependences of the coercive force, nucleation and annihilation fields were also analyzed. The main results are correct and sound interesting. And the paper is well written. In my opinion, the manuscript can be published.

Author Response

Thanks to the Referees for their valuables and useful comments and criticisms to our manuscript. In what follows we answer to every comment and suggestion pointed out for everyone:

Reviewer #1:

In the paper, the authors present a study on the vortices formation in iron quadrangular nanoprisms and their stability under the influence of a unidirectional and uniform in-plane applied field, in which different aspect ratios of a square base prism of thickness with free boundary conditions were considered. The aspect ratio dependences of the coercive force, nucleation and annihilation fields were also analyzed. The main results are correct and sound interesting. And the paper is well written. In my opinion, the manuscript can be published.

Answer:

Thank you very much to the Referee #1 for the comments.

Reviewer 2 Report

The Authors presents nice results related with the zero temperature hysteretic properties of iron (Fe) quadrangular nanoprisms and the size conditions underlying magnetic vortex states formation. They took into account different aspect ratios of a square base prism of thickness t with free boundary conditions in order to summarize their results in a proposal of a feld-driven magnetic phase diagram where such vortex states are stable along the hysteresis loops.

Questions to Authors:

1. Text of the manuscript in line 131 – 134 in relation to Figure 3 – in both cases in Fig. 3d) and 3e) core of the vortex is below y=0? Is it correct?

2. Could author explain behavior visible at Figure 3 presenting magnetic configurations for three different thicknesses, namely t = {42, 45, 48} nm and different values of magnetic field? For t= 42 nm at Fig. 3 a) “max” or “center-vortex” is on the top of the image, then decreasing the field “center - vortex” go below the y=0 for H=25 mT. For H=0 mT “center-vortex” lying at y=0 and again go below y=0 for H= - 25 mT. Additionally one may observe changes of domain color when field is changing from 25 mT to 0 or -25 mT. This behavior is different in relation to t= 45 (48) nm where changing field “max”/”center vortex” has “linear” change – field is decreasing the “max”/”center vortex” traveling from bottom (well below/below y=0) of the image to top (above y=0) of the image. Also no domain color changes is observed?

When we decrease the field “center” Especially column corresponding to t= 42 nm and their behavior in different H.  

3. I also propose to read the manuscript carefully and check vocabulary of English words, like e.g. “sistematically” in line 66; “desmagnetizing energy (J)” in label axis description in Figure 4; “thikcnesses”…

Generally, manuscript is well organised however it needs additional revision.

Generally manuscript is well organized however it needs additional revision.

Author Response

Thanks to the Referees for their valuables and useful comments and criticisms to our manuscript. In what follows we answer to every comment and suggestion pointed out for everyone:

Reviewer #2:

The Authors presents nice results related with the zero temperature hysteretic properties of iron (Fe) quadrangular nanoprisms and the size conditions underlying magnetic vortex states formation. They took into account different aspect ratios of a square base prism of thickness t with free boundary conditions in order to summarize their results in a proposal of a feld-driven magnetic phase diagram where such vortex states are stable along the hysteresis loops.

Questions to Authors:

  1. Text of the manuscript in line 131 – 134 in relation to Figure 3 – in both cases in Fig. 3d) and 3e) core of the vortex is below y=0? Is it correct?

Answer:

No, it is not correct. Effectively we made an editing mistake where figure 3d resulted rotated, which in the new version of the manuscript is now corrected. In particular, for this figure, vortex core is above y=0 in correspondence with the asymmetry of the initial majority region or domain of the magnetic moments shown in figure 3a. We acknowledge to the Referee for this acute observation. Moreover, the snapshots shown in figure 3 correspond to L=30 nm and not L=300 nm. Nevertheless, this typo does not affect the analysis which is similar for other aspect ratios (see line 137) and caption of figure 3.

  1. Could author explain behavior visible at Figure 3 presenting magnetic configurations for three different thicknesses, namely t = {42, 45, 48} nm and different values of magnetic field? For t= 42 nm at Fig. 3 a) “max” or “center-vortex” is on the top of the image, then decreasing the field “center - vortex” go below the y=0 for H=25 mT. For H=0 mT “center-vortex” lying at y=0 and again go below y=0 for H= - 25 mT. Additionally one may observe changes of domain color when field is changing from 25 mT to 0 or -25 mT. This behavior is different in relation to t= 45 (48) nm where changing field “max”/”center vortex” has “linear” change – field is decreasing the “max”/”center vortex” traveling from bottom (well below/below y=0) of the image to top (above y=0) of the image. Also no domain color changes is observed?

When we decrease the field “center” Especially column corresponding to t= 42 nm and their behavior in different H.

Answer:

Referee is completely right. As we mentioned above in the answer of point 1, we made a mistake while editing this figure. In the new version of the manuscript this particular feature is now corrected, where vortex core is above y=0 in figure 3d, and the correspondence in the color of the domains among the figures is now consistent.

  1. I also propose to read the manuscript carefully and check vocabulary of English words, like e.g. “sistematically” in line 66; “desmagnetizing energy (J)” in label axis description in Figure 4; “thikcnesses”…

Answer:

Referee is right. The English has been revised again throughout the entire manuscript including the words pointed out by the Referee.

Generally, manuscript is well organised however it needs additional revision.

Answer:

Thanks to the Referee once more.

Reviewer 3 Report

The authors systematically investigated the hysteric properties of Fe quadrangular nanometric prisms and vortex stability at T = 0 K. Main conclusion is summarized in Fig. 8 and they have proposed field-driven magnetic phase diagram for Fe nanometric square prisms.

Relationship between Fig. 8 and other figures seems unclear. Figure 8 presents four kinds of magnetic phases in the (t, L) plane but there is no axis related to magnetic field in spite of “field-driven”. More explanation might be needed for readers how Fig. 8 was obtained from their discussion. Please define t1c, t2c, t3c, FM1, FM2, V1, and V2.

Author Response

Thanks to the Referees for their valuables and useful comments and criticisms to our manuscript. In what follows we answer to every comment and suggestion pointed out for everyone:

Reviewer #3:

The  Comments and Suggestions for Authors

The authors systematically investigated the hysteric properties of Fe quadrangular nanometric prisms and vortex stability at T = 0 K. Main conclusion is summarized in Fig. 8 and they have proposed field-driven magnetic phase diagram for Fe nanometric square prisms.

Relationship between Fig. 8 and other figures seems unclear. Figure 8 presents four kinds of magnetic phases in the (tL) plane but there is no axis related to magnetic field in spite of “field-driven”. More explanation might be needed for readers how Fig. 8 was obtained from their discussion. Please define t1ct2ct3cFM1FM2V1, and V2.

Answer:

Thanks to the Referee for the comments.

Even though there is no axis related to the magnetic field in the (t,L) diagram in figure 8, we want to stress that such a diagram was constructed by considering the shape and characteristics of the hysteresis loops for the different (t,L) values studied. It is in that sense that we call the diagram a “field-driven magnetic phase diagram” in order to highlight the difference with what would be a zero-field diagram.

Nevertheless, we agree with the Referee that more information is needed on how the diagram was obtained as well as a clear definition of labels t1ct2ct3cFM1FM2V1, and V2 . To do this, paragraphs on page 4 below figure 1 (lines 100 to 128 in the new version) and last paragraph on page 7 (lines 157 to 166 in the new version) were modified. Labels and captions of figures 2 and 5 were also modified for clarification.

Round 2

Reviewer 3 Report

Fisrt draft was properly modified and therefore explanation of Fig. 8 in second draft becomes better description. In my opinion, present form (second draft) can be published.